# New Insights into the Communications of the Facial Vein with the Dural Venous Sinuses

**DOI:** 10.3390/medicina59030609

**Published:** 2023-03-19

**Authors:** Martin Siwetz, Hannes Widni-Pajank, Niels Hammer, Ulrike Pilsl, Simon Bruneder, Andreas Wree, Veronica Antipova

**Affiliations:** 1Division of Macroscopic and Clinical Anatomy, Gottfried Schatz Research Center, Medical University of Graz, Auenbruggerplatz 25, A-8036 Graz, Austria; 2Department of Oral and Maxillofacial Surgery, Klagenfurt Am Wörthersee Clinic, Feschnigstraße 11, A-9020 Klagenfurt am Wörthersee, Austria; 3Department of Orthopedic and Trauma Surgery, University of Leipzig, D-04103 Leipzig, Germany; 4Division of Biomechatronics, Fraunhofer Institute for Machine Tools and Forming Technology Dresden, D-09126 Dresden, Germany; 5Department of Oral and Maxillofacial Surgery, Medical University of Graz, Auenbruggerplatz 5, A-8036 Graz, Austria; 6Institute of Anatomy, Rostock University Medical Center, Gertrudenstr. 9, D-18057 Rostock, Germany

**Keywords:** facial vein, angular vein, cavernous sinus, anatomical variations, anastomoses, extracranial and intracranial venous system

## Abstract

*Background and Objectives*: Anastomoses of the extracranial and intracranial venous system have been described in the literature. The presence of such anastomoses may facilitate a possible spread of infection into the dural venous sinuses. However, the frequency and relevance of such anastomoses is highly debated. The aim of this study was to quantify frequencies of anastomoses between the facial vein and the dural venous sinuses. *Materials and Methods*: In 32 sides of 16 specimens, latex was injected into the facial vein. Dissection was carried out to follow and described these anastomoses, yielding the presence of latex in the intracranial venous system. *Results*: In 97% of cases, a dispersal of latex into the cavernous sinus as well as anastomoses was observed. A further dispersal of latex into other dural venous sinuses was found at rates ranging between 34% (transverse sinus)—88% (superior petrosal sinus), respectively. *Conclusions*: The presence of anastomoses between the extracranial and intracranial venous system in a majority of cases needs to be considered when dealing with pathologies as well as procedures in the facial region.

## 1. Introduction

The facial vein is typically described as taking a course from the medial angle of the eye towards to the lower margin of the mandible [1,2,3,4]. Its uppermost and most distal portion, the angular vein, is typically described to form anastomoses with the ophthalmic vein via the supratrochlear and supraorbital veins [5,6]. Furthermore, an anastomosis of the extracranial and intracranial venous system can be described via the pterygoid plexus [7,8]. The direction of drainage of the facial vein as well as the presence of valves determining their direction of drainage are ongoingly debated in the literature with some authors describing the main direction of drainage as being towards the intracranial dural venous sinuses, and others describing a directed blood flow towards the facial vein [6,9,10,11]. With an increasing number of aesthetic procedures performed in the facial region including filler and Botulinum toxin injections [12,13], a number of complications may arise as a consequence of such procedures. These include local infections, thrombosis and skin necrosis as well as a spread of infections or thrombosis into the intracranial blood vessels [14,15,16,17]. A spread of infection does not only occur iatrogenically as a complication of an aesthetic procedure but also as a complication of local infection such as furuncles, carbuncles, canine fossa abscesses or periorbital cellulitis [18,19,20]. A spread of infection from the facial regions into the dural venous sinuses may result in septic cavernous sinus thrombosis [18,21,22]. Septic cavernous sinus thrombosis is a life-threatening condition, with high morbidity and mortality rates being described at around 30% [18,19,20]. In aesthetic filler therapy, a rare but serious complication is intravascular application leading to thrombosis or embolization [13,23].

Based on the possibility of severe complications and the controversy in the literature on the communication of the facial veins with the dural venous sinuses, this study aimed at quantifying the rate of communication of the dural venous sinuses with the facial venous system. This morphological data may help understand the possible complications of a spread of infection but also to quantify the potential risks following medical intervention in the region of the medial eye angle.

## 2. Materials and Methods

For this study, 16 whole heads of anatomic specimens were used, including 8 males and 8 females (the age range at the time of death was 39–92 years, with a mean age of 77.9 years, a mean body height of 171.4 cm, and a mean body weight of 74.3 kg), with 16 right and 16 left sides. The specimens were all embalmed using the modified Thiel technique [24,25,26], and the arterial system was injected using a radiopaque red latex mass via the common carotid artery [25]. All specimens were sent to the Division of Macroscopic and Clinical Anatomy of the Medical University of Graz (Graz, Austria) under the approval of the Anatomical Donation Program of the Medical University of Graz (Graz, Austria) and in accordance with the Austrian laws concerning body donations. While alive, all the body donors had given their informed consent for the donation of their post mortem tissues for research purposes.

Specimens were only included into this study if they showed no pathological lesions in the face, eye, and neurocranial regions and no signs and history of surgery or tumors in the region. For the systematic visualization and examination of the communication between the facial vein and the dural venous sinuses, the facial vein was injected with blue-colored latex. Therefore, the main trunk of the facial vein was identified at the base of the mandible. It was ligated proximally so that no retrograde flow of injection mass could occur and was injected using an injection mass consisting of 70% distilled water and 30% nature-latex GIVUL MR (Fa. Helmut Bergk, Frankfurt am Main, Germany). This injection mass was mixed with blue dye (“Abtönfarbe Pintasol Blau E-WL5”, Fa. Helmut Bergk). After the injection and the curing of the latex, the calvaria was opened and the brain was removed. The dural venous sinuses were incised systematically and inspected for blue coloration.

## 3. Results

In all the 16 specimens (both sides in 15 specimens and one side in one specimen), an inflow of the colored latex into the dural venous sinuses was present. This inflow confirmed the presence of anastomoses between the facial vein and the dural venous sinuses via the angular vein (Figure 1a,b). In one specimen, the right-sided anastomosis between the angular vein and the ophthalmic vein was rather weak, leading to only slight coloration of the dural venous sinuses on this side.

Overall, in 97 % of cases (31/32), a dispersal of the latex into the cavernous sinus was observed. When further investigating the dural venous sinuses (Figure 2a,b), a dispersal of the latex was found in the anterior intercavernous sinus in 50% (8/16) of cases, and into the posterior intercavernous sinus in 81% (13/16) of cases. Further dispersal via the superior petrosal sinus was found in 88% (28/32) and via the inferior petrosal sinus in 78% (25/32) of cases. A connection of the cavernous sinus (Figure 3) to the basilar plexus was present in 69% (11/16) of cases, and dispersal into the transverse sinus was present in 34% (11/32) of cases.

## 4. Discussion

The aim of the study was to quantify the rate of communication of the facial venous system with the dural venous sinuses, since a spread of infection or thrombosis through this pathway may cause life-threating complications.

The presence of venous valves and the direction of blood flow in the facial region remains a matter of ongoing debate in the literature of this subject. Anatomy textbooks commonly report that facial veins are valveless—making it easier for infections to spread from the “triangle of danger”—being marked out by the angle of the mouth and the root of the nose [27] to the intracranial venous sinuses [28,29,30]. Some studies suggest that the facial vein does not differ from other veins and thus does have valves for the direction of blood flow [6,11], at least in the frontal region and the region of the glabella. Other studies found no valves in the upper portion (along the nose) of the facial veins [10,31]. An absence of venous valves could further support the spread of infection and/or the formation of cavernous sinus thrombosis. These findings are further backed by the findings of Zhang and Stringer [6], who found venous valves in the superior orbital veins in 75% of cases, as well as in the majority of facial and angular veins, but none in the inferior ophthalmic veins. 

The authors also concluded that this was not a matter of the absence of venous valves but the existence of anastomoses between the facial vein and cavernous sinus, and the direction of blood flow that are important in the spread of infection via the face [6,28].

The formation of septic cavernous sinus thrombosis is caused by a spread of infection into the cavernous sinus. This can either be due to a direct extension of sinusitis, but also due to the indirect seeding of an infection via the bloodstream [20]. As our findings showed that there was a connection between the facial vein and the cavernous sinus in almost all cases (97%), an anatomical route for potential infections spreading into the cavernous sinus seems most commonly present.

The here-stated morphological findings have not assessed the presence of venous valves in the facial veins and its branches; however, a connection of the extracranial venous system with the intracranial venous system was found in 100% of specimens.

Regarding the high rate of communication between the facial venous system and the dural sinuses confirmed in this study, special care should be taken in clinical practice to prevent the spread of facial infection or thrombosis into the dural sinuses.

## 5. Conclusions

In 97% of cases, a connection between the outer facial veins and the intracranial dural sinuses was present, therefore leading to a potential risk of facial infections spreading into the dural venous sinuses.

## 6. Limitations

The study was performed only in embalmed specimens, with fixatives resulting in tissue distortion events such as cellularization and degreasing [22]. Based on these effects, as well as the effects of post-mortem delay and of the design of the study, no final conclusion on the presence of functional valves could be deduced. Further studies on unfixed tissues along with histological workup need to be performed to prove the presence or absence of valves. Furthermore, the number of specimens included in this study was limited.

## Figures and Tables

**Figure 1 medicina-59-00609-f001:**
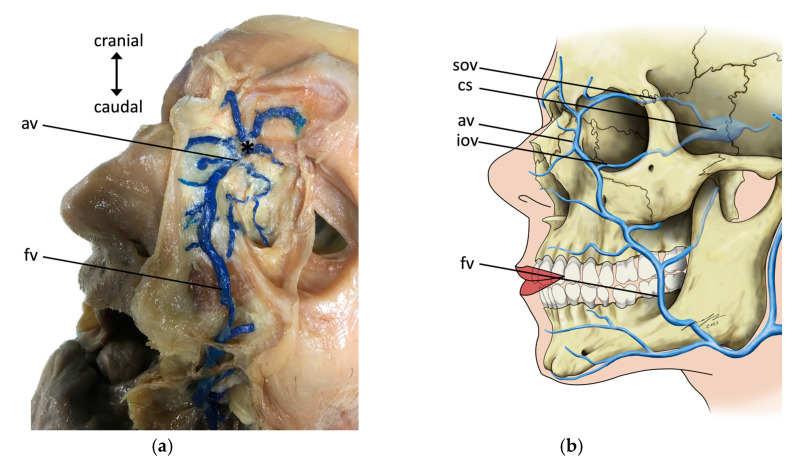
Illustration of the anastomosis (*) of the facial vein (fv) and the superior ophthalmic vein (sov). (**a**) Left side of a male specimen’s face. The facial vein (fv) was injected with blue latex and carefully dissected. Anastomosis via the angular vein (av) was found. (**b**) A schematic overview of the anastomoses between the facial vein and the superior ophthalmic vein as well as the intracranial dural sinuses. cs—cavernous sinus, and iov—inferior ophthalmic vein.

**Figure 2 medicina-59-00609-f002:**
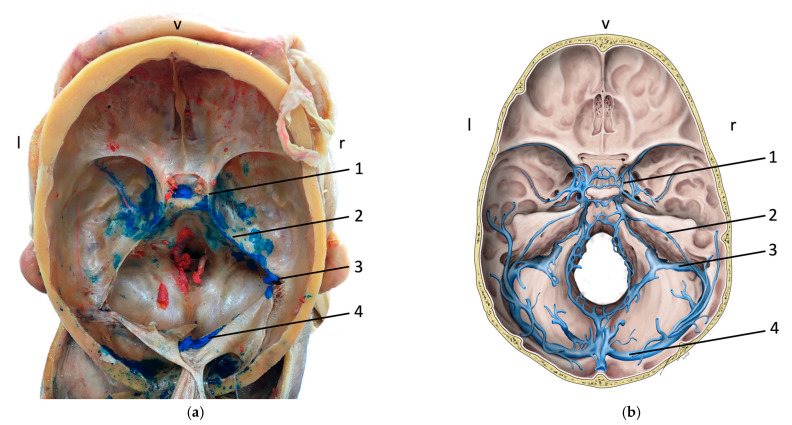
Illustration of the intracranial aspect of the dural sinuses. (**a**) A specimen’s intracranial aspect after removal of the calvaria and the brain. The dispersion of the latex injected via the facial vein into the (1) cavernous sinus, (2) the superior petrosal sinus, (3) the sigmoidal sinus, and (4) the transverse sinus is indicated. (**b**) A schematic overview over the intracranial dural sinuses. (1) cavernous sinus, (2) the superior petrosal sinus, (3) the sigmoidal sinus, and (4) transverse sinus are indicated; v—ventral, l—left, and r—right.

**Figure 3 medicina-59-00609-f003:**
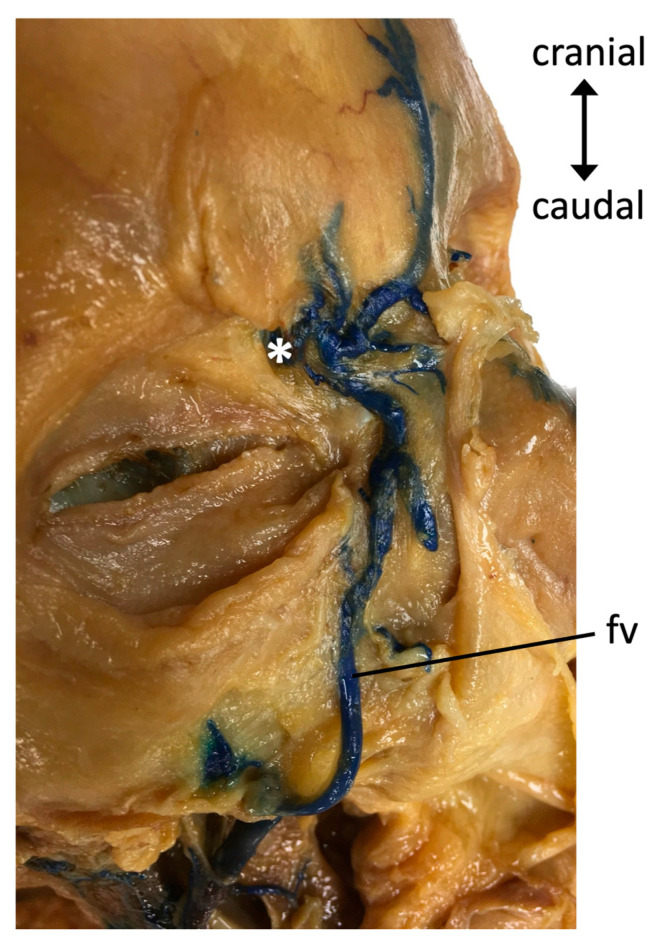
The right side of a male specimen’s face. The facial vein (fv) was injected with blue latex and carefully dissected. The anastomosis between the facial vein and the superior ophthalmic vein (*) can be seen.

## Data Availability

Not applicable.

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
