# Peer review of "New Insights into the Communications of the Facial Vein with the Dural Venous Sinuses"

_medicina, 2023, doi:10.3390/medicina59030609_

Round 1
Reviewer 1 Report
The authors present a morphological study about the anastomosis of the facial vein with the dural sinus system. Using latex injection, the authors could demonstrate a connection of the facial vein with the cavernous sinus in 97 % of cases (31/32). This anatomical study is of interest for both anatomists and clinicians in the field of aesthetic medicine. The dissections were performed carefully, but some minor points have to be addressed prior to publication.
- In figure 2 red colored latex injection in the arterial system is visible. Unfortunately, this is not explained in the text.
- The authors only discuss the anastomosis between the facial vein and the sinus system via superior ophthalmic vein. Please discuss possible alternative routes.
- Did you observe sex or age specific differences in the distribution of the colored latex?
Author Response
Dear Members of the Editorial Board,
Dear Reviewers,
Dear Prof. Varga,
Dear Prof. Kachlík,
Please find attached our revised article titled “New insights into the communications of the facial vein with the dural venous sinuses”. We thank you very much to consider our article for publication. In the following all reviewer quests and comments are answered. All revisions are marked up using the “Track Changes” function in the revised version of the manuscript.
Reviewer 1:
Thank you very much for your input and your kind comments on our manuscript.
- In figure 2 red colored latex injection in the arterial system is visible. Unfortunately, this is not explained in the text.
Reply: The Materials and Methods section has been altered accordingly, so that the injection of red latex, which is a commonly used add-on to our standard embalming protocol, is now mentioned (page 2, lines 71-72). Furthermore, figure 2 has been altered so that there is 2a and 2b with 2b being a schematic overview over the intracranial dural sinuses (page 4, lines 114-121).
- The authors only discuss the anastomosis between the facial vein and the sinus system via superior ophthalmic vein. Please discuss possible alternative routes.
Reply: We thank you for your comment. The theoretically possible course via the pterygoid plexus is now discussed (page 1, lines 43-45).
- Did you observe sex or age specific differences in the distribution of the colored latex?
Reply: Thank you for your question. Since the main focus of this work was to explore the connections between the facial vein and the dural sinus, and this connection was found in every individual at least on one side, no sex specific comparisons on the distribution of the latex were performed.
Reviewer 2
Thank you very much for your input and your kind comments on our manuscript.
-The manuscript entitled: "New insights into the communications of the facial vein with the dural venous sinuses" is an interesting work due to its clinical importance, since it confirms the constant communication between the facial vein and the intracranial dural sinuses. However, the authors have not been able to confirm the presence of valves, perhaps they should add that with the technique used it has not been possible to identify valves.
Reply: The limitations section has been altered to further clarify that it cannot be deduced form our findings whether functional venous valves are present and that further research needs to be performed to answer this question (page 6, lines 166-168).
-Figure 2 is not clear what intracranial dural sinuses are filled with latex and therefore participate in communication with the facial vein. Perhaps a simple diagram of the sinuses could clarify this aspect for readers.
Reply: Thank you for your input. Another figure as a schematic depiction of the dural sinuses has been added analogue to figure 1 a and b to further clarify our anatomical findings (page 4, lines 114-121).
Finally, we appreciate your comments on improving our manuscript and
figures. We were glad to make these clarifications according to your
valuable comments. If there is anything left, please do not hesitate to contact us.
Sincerely on behalf of all authors,
Veronica Antipova

Reviewer 2 Report
The manuscript entitled: "New insights into the communications of the facial vein with the dural venous sinuses" is an interesting work due to its clinical importance, since it confirms the constant communication between the facial vein and the intracranial dural sinuses. However, the authors have not been able to confirm the presence of valves, perhaps they should add that with the technique used it has not been possible to identify valves.
Figure 2 is not clear what intracranial dural sinuses are filled with latex and therefore participate in communication with the facial vein. Perhaps a simple diagram of the sinuses could clarify this aspect for readers.
Author Response

(The authors gave the same response as above.)
